# Neural scaling laws for phenotypic drug discovery

## Abstract

Recent breakthroughs by deep neural networks (DNNs) in natural language processing (NLP) and computer vision have been driven by a scale-up of models and data rather than the discovery of novel computing paradigms. Here, we investigate if scale can have a similar impact for models designed to aid small molecule drug discovery. We address this question through a large-scale and systematic analysis of how DNN size, data diet, and learning routines interact to impact accuracy on our Phenotypic Chemistry Arena (*Pheno-CA*) benchmark — a diverse set of drug discovery tasks posed on image-based high content screening data. Surprisingly, we find that DNNs explicitly supervised to solve tasks in the *Pheno-CA* do not continuously improve as their data and model size is scaled-up. To address this issue, we introduce a novel precursor task, the *Inverse Biological Process* (IBP), which is designed to resemble the causal objective functions that have proven successful for NLP. We indeed find that DNNs first trained with IBP then probed for performance on the *Pheno-CA* significantly outperform task-supervised DNNs. More importantly, the performance of these IBP-trained DNNs monotonically improves with data and model scale. Our findings reveal that the DNN ingredients needed to accurately solve small molecule drug discovery tasks are already in our hands, and project how much more experimental data is needed to achieve any desired level of improvement. We release our Pheno-CA benchmark and code to encourage further study of *neural scaling laws* for small molecule drug discovery.

## 1 Introduction

Rich Sutton (Sutton, 2019) famously wrote, "the biggest lesson that can be read from 70 years of AI research is that general methods that leverage computation are ultimately the most effective, and by a large margin." The scale of compute, model, and data have proven over recent years to be the most important factors for developing high performing systems in nearly every domain of AI including computer vision (Dehghani et al., 2023), natural language processing (Kaplan et al., 2020), reinforcement learning (Hilton et al., 2023), protein folding (Lin et al., 2023), and design (Hesslow et al., 2022). The foundation of each of these revolutions-of-scale rests on empirically derived "neural scaling laws," which indicate that continued improvement on a given domain's tasks are constrained by compute, model, and data scale rather than novel algorithmic solutions or additional domain-knowledge. Thus, one of the extraordinary opportunities for AI is finding and exploiting similar scaling laws in domains that have not benefited from them yet.

Small molecule drug discovery is one of the domains where scaling laws could have an outsized impact. Biological experiments are costly and time intensive, while the space of molecules has been estimated to contain as many as $10^{60}$ compounds with drug-like properties (Lipinski et al., 2012). The current standard approach for identifying interactions between small molecules and biological targets involves high throughput screening (HTS), in which libraries of hundreds of thousands of molecules are tested empirically in parallel for specific biological readouts at great cost. The ability to accurately predict *in silico* whether a small molecule engages a biological target would at the very least reduce the size of the chemical libraries needed to find bioactive molecules and support significantly faster and cheaper discovery. Moreover, if models for small molecule drug discovery follow similar scaling laws as those discovered for natural language processing (Kaplan et al., 2020), then it would mean that even the loftiest goals may be within reach, such as screening larger parts of the $10^{60}$ space of molecules to find treatments for today's intractable diseases. Can DNNs speed up drug discovery, and if so, do their abilities follow neural scaling laws?

One of the most promising avenues for generating data that could be used to train DNNs on drug discovery tasks is image-based high-content screening (iHCS). This type of screen is widely used to measure the effects

of drugs and find targets for treating disease because it is can capture a large variety of biological signatures through different stains or biosensors, and has been helpful in drug discovery applications including hit identification (Simm et al., 2018; Bray et al., 2017) and expansion (Hughes et al., 2020), lead optimization (Caie et al., 2010), generating hypotheses on a drug's mechanism-of-action (Young et al., 2008; Boyd et al., 2020; Sundaramurthy et al., 2014) and target (Schenone et al., 2013), and also identifying and validating disease targets (see Chandrasekaran et al., 2021 for a review). While iHCS is potentially more flexible than standard biochemical assays used in drug discovery, it still requires significant time, money, and effort to set up and run. The recently released JUMP dataset (Chandrasekaran et al., 2023a;b) contains nearly two orders of magnitude more iHCS data than was previously available to the public (Bray et al., 2017), and therefore represents a significant opportunity for deep learning. However, it is still unclear if DNNs can leverage the data in JUMP for drug discovery.

Here, we use the JUMP dataset to investigate if DNNs trained on it for small molecule drug discovery tasks follow neural scaling laws. A positive answer to this question could bring about a revolution in biomedicine that mimics the ones in natural language processing and computer vision over recent years, making it faster, easier, and cheaper than ever to discover drugs.

**Contributions.** We began by augmenting the JUMP dataset with our Phenotypic Chemistry Arena (*Pheno-CA*): a diverse set of drug discovery tasks posed on a subset of images in JUMP. We then tested if the performance of DNNs trained to solve each task could be predicted by the size of their models or the amount of data they were trained with. Surprisingly, it could not: the performance of these "task-supervised" DNNs was either unaffected or hurt by an increase in data and model sizes (Fig. A1a). However, DNNs in domains like natural language processing and vision rely on specific objective functions to achieve efficient scaling — for instance, GPT models use the causal language modeling objective (Kaplan et al., 2020). We reasoned that a similar precursor task, especially one that could force DNNs to learn a causal model of biology, could have have a large impact on scaling. We therefore developed a novel precursor task, the *inverse biological process* (IBP), and performed large-scale and systematic experiments on the *Pheno-CA* to understand how this task interacted with the size of DNN architectures and the amount of data used in training them. Through this large-scale survey, we found the following:

- DNNs pretrained with IBP significantly outperform task-supervised DNNs on the *Pheno-CA*.

- DNNs pretrained with IBP also follow linear scaling laws on the *Pheno-CA* that accurately predict how many novel samples and replicates are needed to achieve arbitrary levels of accuracy.

- IBP-trained DNNs improved in predictable ways as the total number of model parameters was increased. The effect of model depth, on the other hand, was less clear, and impacted only a subset of tasks.

- Scaling laws on IBP-trained DNNs indicate that to achieve 100% accuracy on a task like predicting a compound's mechanism-of-action, JUMP would need to be expanded by approximately 3.25M compounds. Achieving this scale of data would take an impossible amount of time and money, meaning that additional experimental advances are needed to improve neural scaling laws and move significantly beyond the performances of our IBP-trained models.

- We release our *Pheno-CA* challenge and code at `https://anonymous.4open.science/r/pub_scaling_mols-B3E1/` to encourage the field to continue investigating scaling laws in iHCS drug discovery.

## 2 METHODS

**JUMP data** The Joint Undertaking for Morphological Profiling (JUMP) project has produced the largest publicly available dataset for iHCS. The dataset consists of images of Human U2OS osteosarcoma cells from 12-different data generating centers. Each image depicts a well of cells in a plate that have been perturbed then stained with the "Cell Painting" toolkit (Bray et al., 2016). Cell Painting involves fixing and staining cells with six dyes that mark eight different cell organelles or compartments: DNA, nucleoli, actin, Golgi apparatus, plasma membrane, endoplasmic reticulum (ER), cytoplasmic RNA, and mitochondria. Together, these stains provide an unbiased read-out on the effects of different perturbations on cell biology (Fig. 1a). JUMP perturbations include the addition of 116,750 different compounds and the knockout of 7,975 genes by Clustered Regularly Interspaced Short Palindromic Repeats (CRISPR) [*]. There are a total of 711,974 compound perturbation images and 51,185 CRISPR perturbation images, which amounts to an average of five replicates of each perturbation type.

---

[*]JUMP also contains gene overexpression manipulations, but we do not include those images in this study.

**(a)**   **Joint undertaking in morphological profiling (JUMP) dataset**

DNA          AGP          ER          RNA          Mitochondria

Concatenated

**(b)**  **Phenotypic chemistry arena**

Test each model on diverse chemistry
tasks posed on fixed test sets.

**Model zoo (1,876 total DNNs)**

▪ Half are IBP-pretrained
▪ Between 10M and 200M parameters
▪ Systematic variations in training data

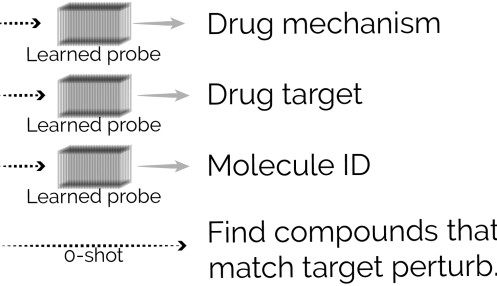

Learned probe → Drug mechanism

Learned probe → Drug target

Learned probe → Molecule ID

0-shot → Find compounds that match target perturb.

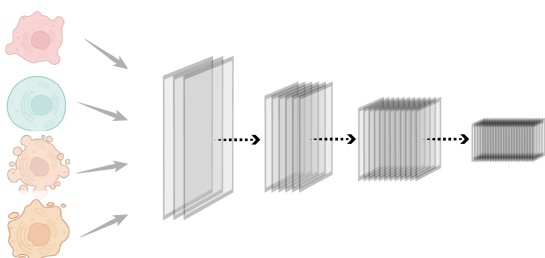

**(c)**   **Forward biological process**

Different molecules yield
different phenotypes in cells.

**Inverse biological process (IBP)**

Learn to predict
compounds from phenotypes.

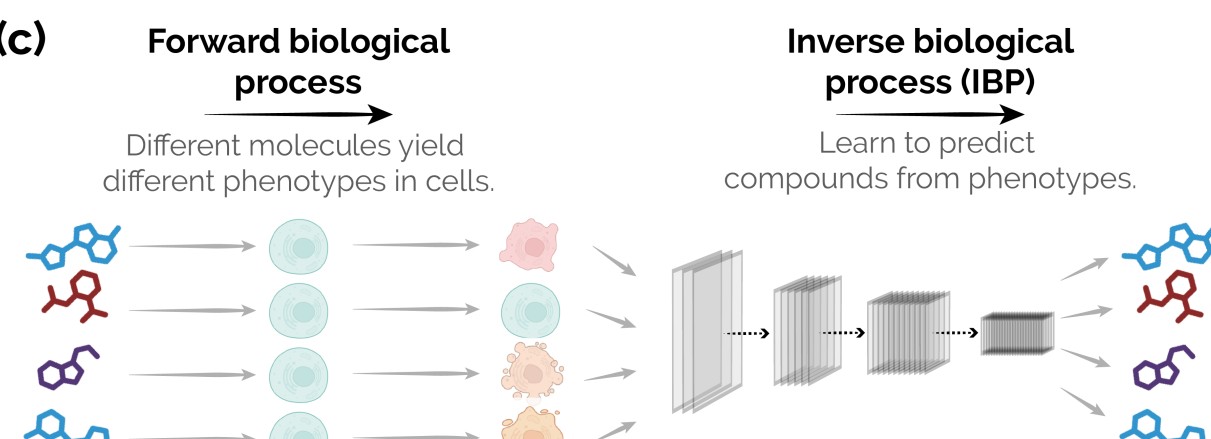

Figure 1: **Designing a scalable precursor task for phenotypic drug discovery. (a)** We investigate the ability of DNNs to learn drug discovery tasks from the large-scale JUMP dataset. **(b)** Our Phenotypic Chemistry Arena (*Pheno-CA*) measures the ability of DNNs trained on JUMP data to solve diverse drug discovery tasks. Task performance is measured through either "learned probes," small neural network readouts that map a DNN's learned representations to the labels for a task, or through "0-shot" evaluations of performance (no task-specific training). **(c)** We find that only those DNNs pretrained on a specialized precursor task — the inverse biological process — follow scaling laws on the *Pheno-CA*.

**Phenotypic Chemistry Arena.**   We introduced the Phenotypic Chemistry Arena (*Pheno-CA*, Fig. 1b) to evaluate DNNs trained on JUMP data for drug discovery tasks The *Pheno-CA* consists of annotations on 6,738 well images from JUMP for four different discovery tasks. These tasks are (*i*) predicting a drug's mechanism-of-action ("MoA deconvolution," 1,282 categories), (*ii*) predicting a drug's target ("target deconvolution," 942 categories), (*iii*) predicting a molecule's identity ("molecule deconvolution," 2,919 categories), and (*iv*)

finding compounds with the same target as a CRISPR perturbation ("compound discovery" Fig. 1b). The remaining images from JUMP are used for training, and depict perturbations from all 116,750 represented in the JUMP dataset (including the 2,919 compounds in the *Pheno-CA*). All well images used in the *Pheno-CA* were held out from model training.

DNN performance on the *Pheno-CA* tasks was measured in different ways. Performance on MoA and target deconvolution was recorded as top-10 classification accuracy, *i.e.*, a model was considered accurate if the model's top-10 predictions included the molecule's true (labeled) MoA or target. Molecule deconvolution performance was recorded using categorical cross entropy loss, which measured how closely the distribution of predicted molecule identities matched the true identity. Finally, to measure how accurately models could find the compounds that match a CRISPR perturbation, we constructed curves that indicated how many guesses it took a model to find the appropriate molecules, then computed area-under-the-curve (AUC).

**Preprocessing**  We preprocessed CP data in two ways. First, we aggregated all cells from a well into a single representation, which captured the effect of its particular experimental perturbation. Second, we normalized these representations to control for experimental nuisances such as well, plate, and batch effects. To aggregate cells into a single well-based representation, we took the median CP-vector per well then normalized these representations by subtracting off the per-plate median representation and dividing by the per-plate inter-quartile-range (Wong et al., 2023). Lastly, before training, we principle components analysis (PCA) whitened the representations (Bell & Sejnowski, 1996), which yielded $878-$dimensional vectors for each well. As we describe in the Appendix C, some of the models were also explicitly trained to ignore experimental nuisances.

**Inverse biological process learning as a generalist precursor task**  DNN and data scale paired with the appropriate precursor training task — so-called causal language modeling — have lead to breakthroughs in natural language processing. Here, we devised a learning procedure that we hypothesized would similarly help DNNs learn biological causality from JUMP data. Our basic insight is that each cell in the JUMP dataset undergoes a "forward biological process", in which the addition of a small molecule transforms its phenotype from a control to a perturbed one (Fig 1c). We reasoned that training a model to invert this process would force it to learn the host of underlying biophysical processes that cause a cell to change phenotypes, and that the resulting model representations would prove useful for downstream discovery tasks including those in the *Pheno-CA* (Ardizzone et al., 2018). We refer to this precursor task as the *inverse biological process* (IBP). If a model improves on tasks in the *Pheno-CA* after IBP-training it means that the motivating hypothesis is at least partially correct. In practice, IBP involves learning to predict a molecule from the phenotype it causes. We investigated the efficacy of IBP on the *Pheno-CA* by first pretraining a DNN for this task before freezing its weights and training task-specific readouts on its representations as detailed below.

**Model zoo.**  We built a large "zoo" of DNNs to understand how changing model architecture, supervision methods, and the amount of data seen during training affects performance on the *Pheno-CA*. Each DNN ended with a task-specific 3-layer multilayer perceptron (MLP), which mapped its representations of image content to a *Pheno-CA* task.

All DNNs consisted of a basic MLP block with a residual connection. The MLP consisted of a linear layer, followed by a 1-D BatchNorm, and finally a gaussian error linear unit (GELU; Hendrycks & Gimpel, 2016). DNNs consisted of 1, 3, 6, 9, or 12 layers of these blocks, each with 128, 256, 512, or 1512 features.

We tested two types of DNN supervision. (*i*) DNNs were directly trained to solve each *Pheno-CA* task. (*ii*) DNNs pretrained with IBP were frozen and their representations were mapped to a *Pheno-CA* task with the 3-layer MLP readout. In other words, we compared DNNs that learned task-specific representations to DNNs that learned IBP representations. Each of these DNNs was also given images from 1e3, 2e4, 5e4, 8e4 or 1e5 molecules that were not included in the *Pheno-CA*, and hence were out-of-distribution (OOD) of that challenge. Our hypothesis was that OOD compound information would help IBP-trained DNNs more accurately model the biophysical effects of compounds on U2OS cells, and ultimately outperform task-supervised DNNs on *Pheno-CA* tasks. Finally, each DNN was trained on 1%, 25%, 50%, 75%, or 100% of replicates of each of the compounds included in their training set.

All combinations of data, model, and supervision parameters yielded 1,876 unique DNNs. Each DNN was implemented in PyTorch and trained using one NVIDIA TITAN X GPU with 24GB of VRAM. DNNs were trained with a AdamW (Loshchilov & Hutter, 2017), a learning rate of 1e-4, a batch size of 6000, and mixed-precision weights using Huggingface Accelerate library (`https://github.com/huggingface/accelerate`).

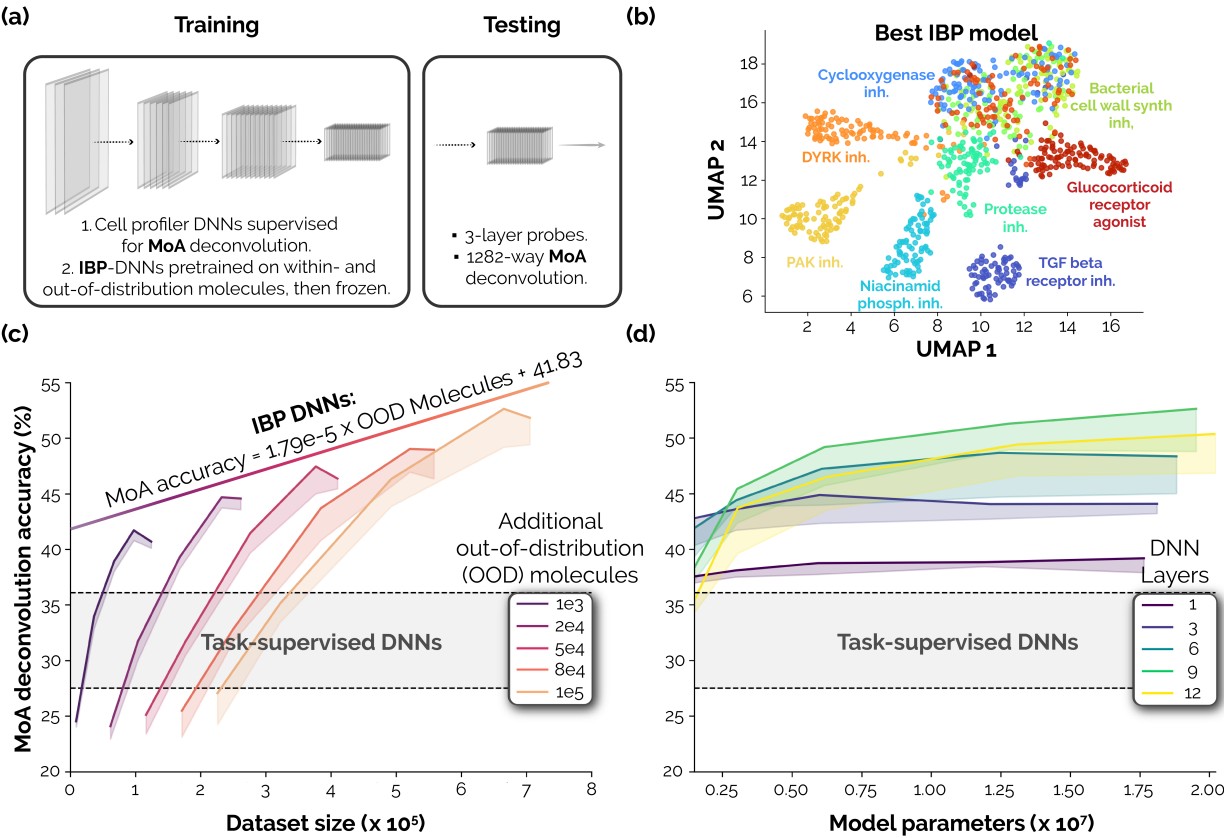

Figure 2: ***Pheno-CA challenge 1***: **Mechanism-of-action (MoA) deconvolution.** (**a**) DNNs were either trained directly for MoA deconvolution from phenotypes or first pretrained on the IBP task before their weights were "frozen" for testing. Testing in each case involved fitting a 3-layer probe to generate MoA predictions for a molecule's imaged phenotype. (**b**) The highest-performing DNN was an IBP-pretrained model. A UMAP decomposition of its image representations qualitatively reveals clustering for the most commonly appearing MoAs. (**c**) IBP-trained DNN performance is a linear function of the amount of data each model is trained on. Each individual colored curves depicts the performance of DNNs trained on a fixed number of molecules that fall "out-of-distribution" of the molecules in the *Pheno-CA*. Decreases on the right end of each curve indicate overfitting. The scaling law depicted here is a linear fit of the max-performing models in each curve. Chance is $7e-2$. (**d**) While DNN performance generally improved as models grew in parameters, 9-layer DNNs were more accurate than 12-layer DNNs.

Training was ended early if test performance stopped improving for 15 epochs. Training took at most 16 hours per DNN.

## 3   RESULTS

The Phenotypic Chemistry Arena (*Pheno-CA*) is a large-scale evaluation benchmark we created to measure the performance of DNNs trained on iHCS for diverse phenotypic drug discovery tasks: (*i*) predicting a drug's mechanism-of-action ("MoA deconvolution"; Chandrasekaran et al., 2021), (*ii*) predicting a drug's target ("target deconvolution"; Schenone et al., 2013), (*iii*) predicting a molecule's identity ("molecule deconvolution"; Chandrasekaran et al., 2021), and (*iv*) finding compounds that have the same target as a CRISPR perturbation (Fig. 1b; Zhang & Gant, 2008; Méndez-Lucio et al., 2020). By surveying 1,876 different DNNs on the *Pheno-CA*, we identified the training routines and DNN architectures that yielded the highest performance on these tasks, and discovered scaling laws that predict performance for certain model classes with respect to the amount and types of data used to train them.

**Challenge 1: Mechanism-of-action deconvolution.** Phenotypic screening is a powerful way to find active compounds in a biologically relevant setting. Phenotypic screens have inspired many important drug programs, including the discoveries of FKBP12 (Harding et al., 1989), calcineurin25 (Liu et al., 1992), and mTOR26 (Brown et al., 1994). However, it often requires substantial effort to understand the mechanism-of-action and targets of small molecules that are bioactive in a phenotypic screen. By MoA, we mean the effect the compound has on a cellular pathway or class of molecules, for instance 'inhibitor of bacterial cell wall synthesis', or 'glucocorticoid receptor agonist'. In contrast, in the target challenge below we refer to the actual cellular component (for instance a specific enzyme) that the compound alters (usually by binding to it). iHCS data has been used in the past to help solve MoA deconvolution through a "guilt-by-association" approach, in which compounds that have known MoAs and targets are added into an experiment and used to deduce those properties in other compounds (Chandrasekaran et al., 2021).

Here, we pose a version of "guilt-by-association" MoA-discovery on JUMP data. Each DNN in our zoo was given images of cells perturbed by different compounds, and trained to predict the MoA of a given compound out of 1,282 possibilities (Fig. 1a). DNNs were either supervised directly for MoA deconvolution or pretrained with IBP (Fig. 2a). Next, DNN weights were frozen and three-layer MLP probes were used to transform image representations from both models into MoA predictions (*i.e.* there was no direct task supervision for IBP models).

Our DNN zoo yielded a wide range of performances on this task. At the low end was a 12.09% accurate 12-layer and 128-feature DNN trained with IBP on 100% of out-of-distribution molecules but only 0.01% of the replicates of each compound. At the high-end was a 52.62% accurate 9-layer and 1512-feature DNN trained with IBP on 100% of out-of-distribution molecules and 75% of the replicates of each compound. The representations of

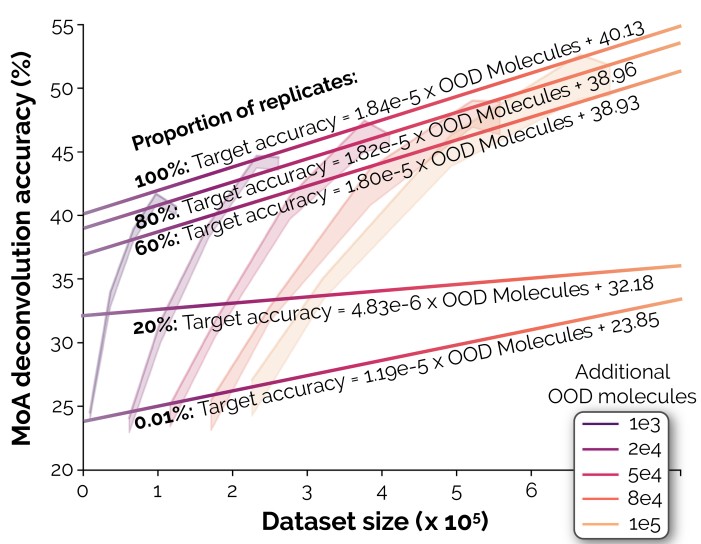

Figure 3: **Experimental replicates improve IBP-trained DNN scaling laws.** DNN performance and scaling laws increased as they were trained with larger amounts of replicates of each experimental perturbation.

this performant IBP-trained DNN clearly separated the phenotypes of different MoAs (Fig. 2b), and it was 46% more accurate than the highest performing task-supervised DNN (36.08%; 6-layer and 256-feature DNN). Overall, IBP-trained DNNs were significantly more accurate at MoA deconvolution than task-trained DNNs ($T(624) = 7.97$, $p < 0.001$). These results indicate that the IBP precursor task promotes generalist representations that outperform task-specific training and are already well suited for MoA deconvolution.

Another key difference we found between task-supervised and IBP-trained DNNs is that the performance of the latter followed a scaling law. The MoA deconvolution accuracy of IBP-trained DNNs linearly increased as they were trained on additional molecules that were *out-of-distribution* (OOD) of the *Pheno-CA* (Fig 2c). The discovered law indicated that IBP-DNN performance increases by 1% with the addition of approximately 56K (non-unique) OOD molecules for training. While DNN performance generally improved with the total number of model parameters, the rate of improvement was higher for 9-layer DNNs than 12-layer DNNs (Fig 2c; analyzed further in Appendix ADD).

We further analyzed scaling laws for MoA prediction by recomputing them for different amounts of experimental replicates. That is, we expected that DNNs which were able to observe more experimental variability would scale better than those that observed less variability. Indeed, we found that more replicates lead to better models on average, and that more data also generally improved the scaling law slope (Fig. 3).

**Challenge 2: Target deconvolution.** Identifying a bioactive molecule's target from its phenotypes is another essential challenge for phenotypic screens. We evaluated the ability of DNNs to automate this task in the *Pheno-CA*, and measured how accurately models can deconvolve a molecule's target from its phenotype.

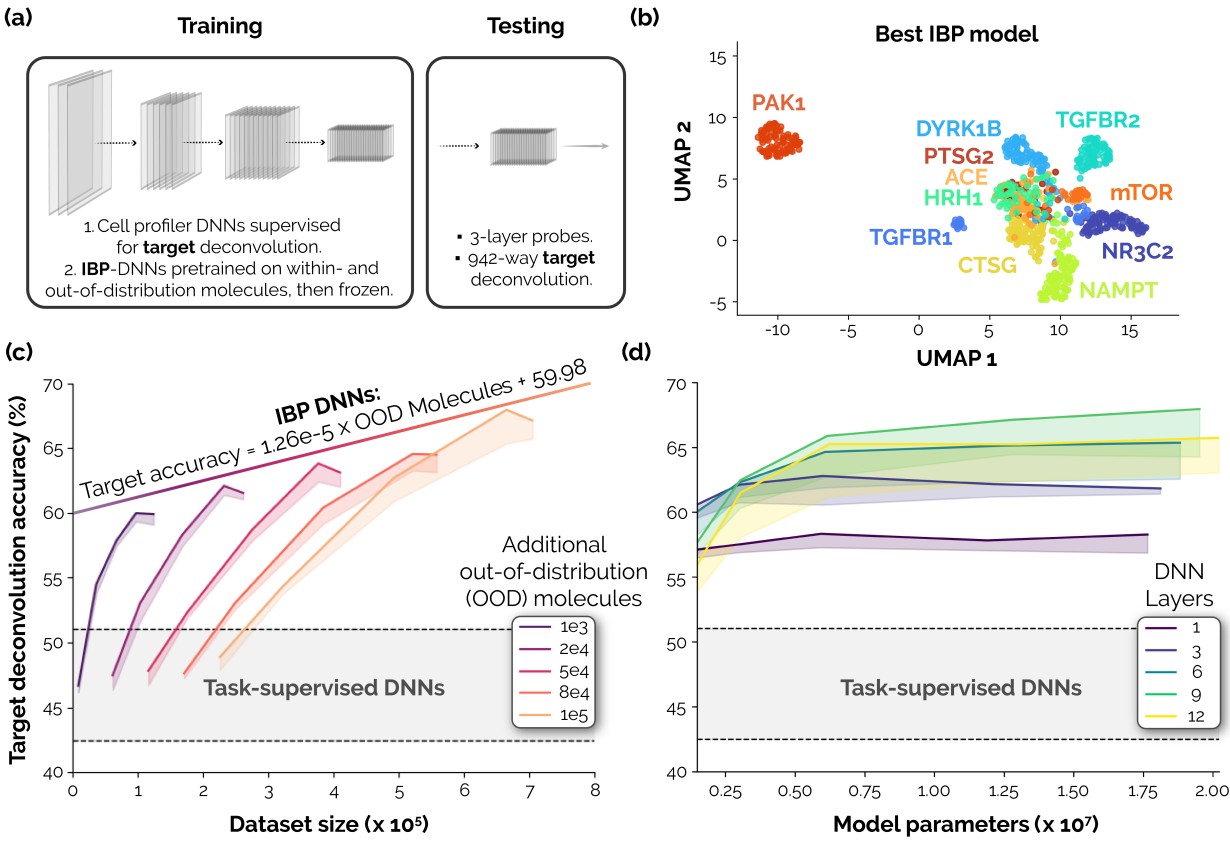

Figure 4: *Pheno-CA challenge 2*: **Target deconvolution.** (**a**) DNNs were either trained directly for target deconvolution from phenotypes or first pretrained on the IBP task then "frozen" for testing. Testing in each case involved fitting a 3-layer probe to generate target predictions for a molecule's imaged phenotype. (**b**) The highest-performing DNN was an IBP-pretrained model, and its representations discriminate between the most commonly appearing targets. (**c**) IBP-trained DNN performance is a linear function of the amount of data each model is trained on. Each individual colored curves depicts the performance of DNNs trained on a fixed number of molecules that fall "out-of-distribution" of the molecules in the *Pheno-CA*. Decreases on the right end of each curve indicate overfitting. The scaling law depicted here is a linear fit of the max-performing models in each curve. Chance is $1.1e-1$. (**d**) While DNN performance generally improved as models grew in parameters, 9-layer DNNs were more accurate than 12-layer DNNs.

This task followed the same general approach as the MoA deconvolution task described above, and tested models for 942-way target deconvolution (Fig. 4a).

As with MoA deconvolution, IBP-trained DNNs were on average significantly better at target deconvolution than task-supervised DNNs ($T(624) = 15.07$, $p < 0.001$). The lowest performing DNN (35.00%) was an IBP-trained 12-layer and 256-feature model trained with 0.01% of out-of-distribution molecules and 25% of the replicates of each compound. The highest performing DNN (67.95%) was an IBP-trained 9-layer and 1512-feature model trained on 100% of out-of-distribution molecules and 75% of the replicates of each compound. The representations of this IBP-trained DNN separated the phenotypes of different targets (Fig. 2b), and it was 33% more accurate than the highest performing task-supervised DNN (51.03%), which was a 6-layer and 512-feature DNN.

IBP-trained DNNs also followed a scaling law on target deconvolution (Fig. 2c). Model performance was linearly predicted by the number of out-of-distribution molecules included in training. The discovered law indicated that IBP-trained DNN performance increases 1% per 79K (non-unique) OOD molecules added to training. As with MoA deconvolution, DNNs improved as they grew in size, but the deepest 12-layer models were again less effective than 9-layer models (Fig. 4d).

**Challenge 3: Molecule deconvolution.** We next probed how well trained DNNs could discriminate between individual molecules by their phenotypes, a task which we call molecule deconvolution. This task involved 2,919-way classification of molecule identities on the *Pheno-CA* (Fig. A2a). In contrast to MoA and target deconvolution, molecular deconvolution represents a distinct challenge since many of the 2,919 compounds may yield quite similar phenotypes. As such, this task measures the ceiling discriminability of phenotypic representations in the models.

The best-performing DNN (4.02 CCE) was an IBP-trained 9-layer and 1512-feature model and trained with 100% of out-of-distribution molecules and 75% of the replicates of each compound. The representations of this IBP-trained DNN separated the phenotypes of different targets (Fig. A2b). IBP-trained DNNs followed another scaling law on this task; their performance was predicted by the number of OOD molecules included in training. The discovered law indicated that IBP-trained DNN performance improved by 1 point of crossentropy loss for every additional 606,061 (non-unique) OOD molecules added into training. DNNs on this task improved as they grew in size, and deeper models performed better (Fig. A2d).

**Challenge 4: Compound discovery.**
Another important use of phenotypic screening is to find compounds that affect biology in ways that resemble a biological perturbation such as a mutation in a specific gene or protein. If we could predict such compounds accurately, we could rapidly assemble compound libraries that we could screen for an illness associated with that mutation.

We investigated the ability of our DNNs to perform this task "zero shot" (*i.e.*, without a task-specific as like in the other challenges) by comparing model representations of molecule phenotypes and of CRISPR manipulations of specific targets. We measured the representational distance between the phenotype of a CRISPR perturbation of one gene and the phenotype of every molecule in the *Pheno-CA*. We then computed the rank-order of the distance between molecules and the target manipulation, and recorded how many molecules one would need to test in order to find those with the same target as the manipulation (Fig. 5). We repeated

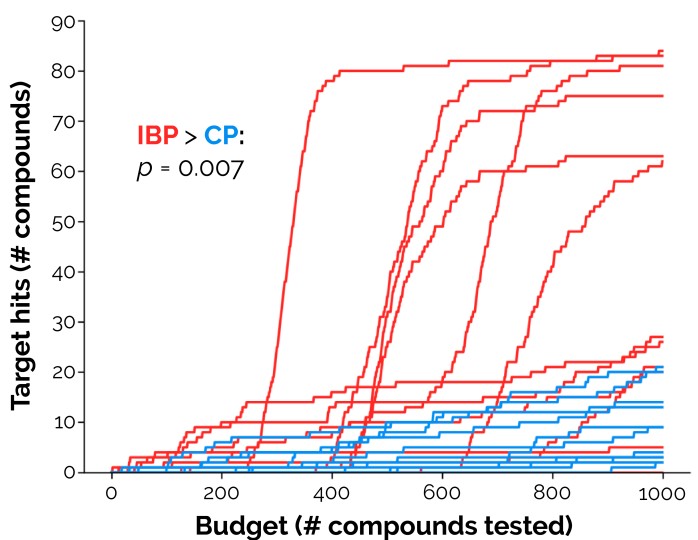

Figure 5: *Pheno-CA challenge 4*: **Compound discovery.** We measured the "zero-shot" effecicay of features from an IBP-trained DNN *vs* cell profiler (CP) for finding compounds that share the target of a CRISPR-perturbation. Lines depict the cumulative number of discovered molecules that match a target. The IBP-trained DNN is significantly better than CP ($p = 0.007$).

this analysis for all compounds with at least 5 replicates in the *Pheno-CA* (12 total) and found that the IBP-trained model with the lowest loss recorded during IBP-training produced representations that were significantly better at task than standard Cell Profiler (CP) representations ($T(11) = 2.91$, $p = 0.007$).

## 4 RELATED WORK

**Deep learning-based chemistry** While computational methods have played a large role in drug discovery since at least the early 80's (Van Drie, 2007), big data and large-scale DNN architectures have recently supported significant advances for a variety of discovery tasks, such as predicting protein folding (Lin et al., 2023) and designing proteins with specific functions (Hesslow et al., 2022). Far less progress has been made in leveraging iHCS data for computational drug discovery tasks, with several notable exceptions. For instance, one study (Wong et al., 2023) found that iHCS data from an earlier and smaller iteration of the JUMP dataset can be used to train models to deconvolve MoAs significantly better than chance. Others have showed that iHCS carries signal for decoding the types of chemical or genetic perturbations that have been applied to cells (Moshkov et al., 2023). Our study takes significant steps beyond these prior ones and aligns iHCS with the goals of large-scale and data-driven AI in three ways: (*i*) we introduce the *Pheno-CA* as a standardized benchmark for model evaluation, (*ii*) we identify model parameterizations that perform well on

this benchmark and are immediately useful for drug discovery, and (*iii*) we discover scaling laws that describe how datasets like JUMP need to be expanded for continued improvements.

**Small molecule drug discovery benchmarks**   While JUMP and the *Pheno-CA* offer a unique opportunity to train and evaluate DNNs on iHCS data for small molecule drug discovery, there are multiple other benchmarks that have focused on structure-based approaches to small molecule design. Fréchet ChemNet Distance (Preuer et al., 2018) (FCD) measures the distance between a model-generated small molecule and the distribution of molecules modeled by a DNN trained to predict the bioactivity of 6,000 molecules. Scoring high on this benchmark means that a model generates compounds that are within distribution of the FCD-model's representations. Guacamol (Brown et al., 2019) and molecular sets (Polykovskiy et al., 2020) (MOSES) are benchmarks that evaluate a model's generated molecules according to their FCD, validity, uniqueness, and novelty. Finally, MoleculeNet (Wu et al., 2017) consists of multiple types of benchmarks for models spanning quantum mechanics, physical chemistry, biophysics, and predictions of physiological properties like toxicity and blood-brain barrier penetration. These benchmarks can synergize with the *Pheno-CA*, for instance, to tune models for filtering molecules predicted by our *Pheno-CA*-adjudicated DNNs to have desirable phenotypic qualities.

## 5  DISCUSSION

The many great breakthroughs of AI over recent years have been guided by the discovery of neural scaling laws (Kaplan et al., 2020; Dehghani et al., 2023). Prior to these scaling laws, it was unclear if achieving human- or superhuman-level performance on challenging tasks in natural language processing and computer vision would require computing breakthroughs that shifted the paradigm beyond deep learning. But scaling laws indicate that sufficiently good algorithms are already in our hands, and that we need more data and compute to unlock their full potential. Here, we provide — to the best of our knowledge — the first evidence that DNNs trained with our IBP precursor task follow similar scaling laws for small molecule discovery.

We find that IBP-trained DNNs are very useful for drug discovery, and significantly better than task-supervised DNNs of any tested size at solving the tasks in our *Pheno-CA*. While the number of experimental replicates included in training affected the overall accuracy of IBP-trained DNNs (Fig. 3), the introduction of additional molecules that fell "out-of-distribution" of the *Pheno-CA* was what actually enabled the accuracy of these models to scale up. This finding implies that the manifold relationship of small molecules and their phenotypes is highly nonlinear and filled with local-minima that make it easy for models to overfit — as if task-supervised DNNs are "looking for their keys under the streetlamp." While it may not be feasible to generate the 14M additional experimental images (3.25M more novel molecules, with about five experimental replicates each) needed to achieve 100% accuracy on a task like MoA deconvolution, continuing to scale experimental data and DNNs towards this goal will unlock extraordinary opportunities to expedite drug discovery for the most challenging diseases we face today. We release our code and the *Pheno-CA* benchmark at `https://anonymous.4open.science/r/pub_scaling_mols-B3E1/` to support these efforts to revolutionize medicine.

**Limitations**   JUMP and other iHCS datasets present significant opportunities for advancing DNNs in small molecule discovery. However, the scaling laws that we discover demonstrate some key limitations. Purchasing just the 3.25M molecules needed to reach 100% accuracy in MoA deconvolution would cost around $325M (assuming $100 per compound). Generating replicate experiments for each could multiply that cost by orders of magnitude. Thus, there is an essential need to identify new imaging and experimental methods that can generate cheaper and better data for training DNNs with IBP. A partial solution to this problem is time-lapse imaging of single cells (Arrasate et al., 2004; Finkbeiner et al., 2015), which enables analysis of single cells over time. Such time-course data has already been successfully used in multiple deep learning applications (Linsley* et al., 2021; Wang et al., 2022; Christiansen et al., 2018) and could approximate and supercharge the beneficial effects of replicates on scaling laws that we observed for MoA deconvolution (Fig. 3).

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

## A  APPENDIX

> **Glossary: Drug discovery**
>
> **Mechanism of action (MoA)**: the biochemical interaction through which a drug substance produces its pharmacological effect.
>
> **Target**: the specific molecular target (protein, RNA, etc.) that a drug interacts with to initiate a biological response.
>
> **Small molecule drug discovery**: the process of identifying and optimizing low molecular weight organic compounds that can modulate biological targets. It involves techniques like high-throughput screening, structure-based drug design, and medicinal chemistry to identify hit compounds, validate their activity, determine their mode of action, and optimize their potency, selectivity, and drug-like properties to generate lead compounds and clinical candidates.

> **Glossary: Machine learning**
>
> **Neural scaling laws:** DNNs trained on certain tasks exhibit predictable improvements as key resources are increased during training and inference. Specifically, metrics like accuracy, loss, and throughput often improve in predictable ways (often power law relationships) as compute, data size, and parameters are increased. These consistent patterns are called scaling laws, and deriving them helps researchers determine optimal resource investments to efficiently maximize model quality for a given task.
>
> **Out-of-distribution data:** these are model inputs that differ from the data used to train a model. This discrepancy can arise when the data is collected under different conditions or from a different environment than the training data. In-distribution data matches the patterns in the training data, while OOD data exhibits new patterns the model has not seen before. Detecting and handling OOD inputs is critical for many applications.
>
> **Precursor tasks:** a common approach in vision and natural language processing is to pretrain on model on a large dataset with a task that could learn generally useful representations of that data. A common approach is with so called self-supervised objetives, which in vision can encourage a model to build up tolerances to transformations that objects undergo in the real world. This pretraining can then improve generalization on downstream classification tasks.

## B  TASK-SUPERVISED DNNS DO NOT EXHIBIT SCALING LAWS

We compared DNNs trained directly on *Pheno-CA* tasks to those that were first pretrained on IBP then frozen and probed (with an MLP) for decisions on *Pheno-CA* tasks. We found that training IBP DNNs on molecules that fell outside the distribution of *Pheno-CA* molecules lead to significant improvements in performance over task-supervised DNNs. Moreover, IBP DNNs followed neural scaling laws; their performance was predicted by the number of additional OOD molecules they were trained on (as well as the number of replicates of each of these molecules). Task-supervised DNNs, on the other hand, do not follow scaling laws (Fig. A1a). Moreover, we found that DNN model scale had very little impact on performance of these models (Fig. A1b).

## C  TRAINING TO IGNORE EXPERIMENTAL NOISE

We developed a novel approach to protect DNNs against the experimental noise, which can overwhelm biological signal captured in iHCS data. To do this, we introduced the source of an image into the input of a model, and added additional losses to the model that made it as inaccurate as possible at predicting the well, batch, or source of an image (see Appendix C for details). These losses were cross entropy between the predicted well/batch/source that an image came from and a uniform distribution across possible wells/batches/sources. Hence, to satisfy this objective, DNNs had to learn representations that were as indiscriminate of wells/batches/sources as possible. We found that DNNs trained with this approach performed significantly better on the *Pheno-CA* than models that were not.

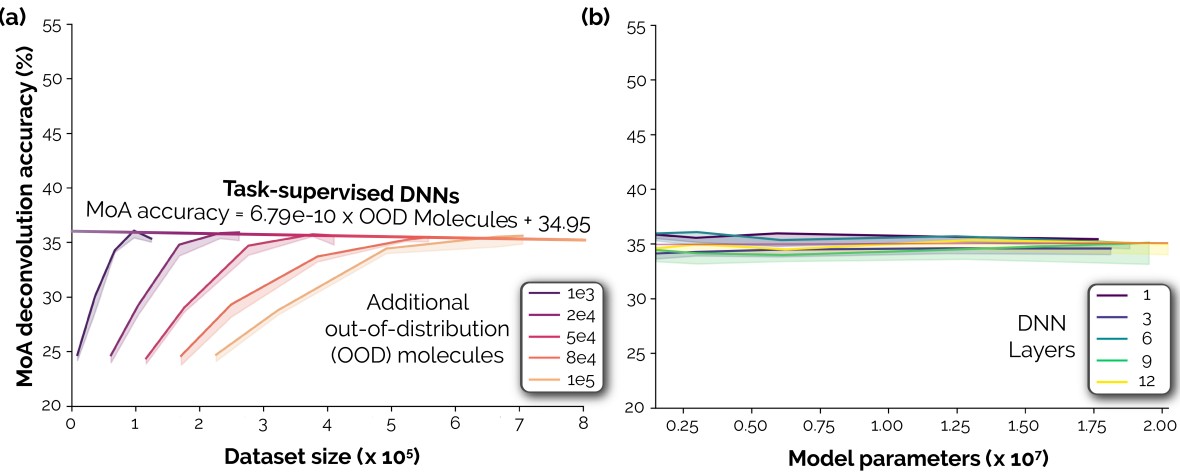

Figure A1: **Task-supervised models trained for mechanism-of-action (MoA) deconvolution do not follow neural scaling laws.** (**a**) The performance of task-supervised DNNs trained for MoA deconvolution as a function of the amount of data they were trained on and the number of additional out-of-distribution (OOD) molecules that were included in training. For each grouping of OOD molecules, DNN performance increased before saturating, and there was no effect of OOD molecules. (**b**) The number of parameters and layers in task-supervised DNNs had a minimal affect on performance.

# D   Challenge 3 results: Molecule deconvolution

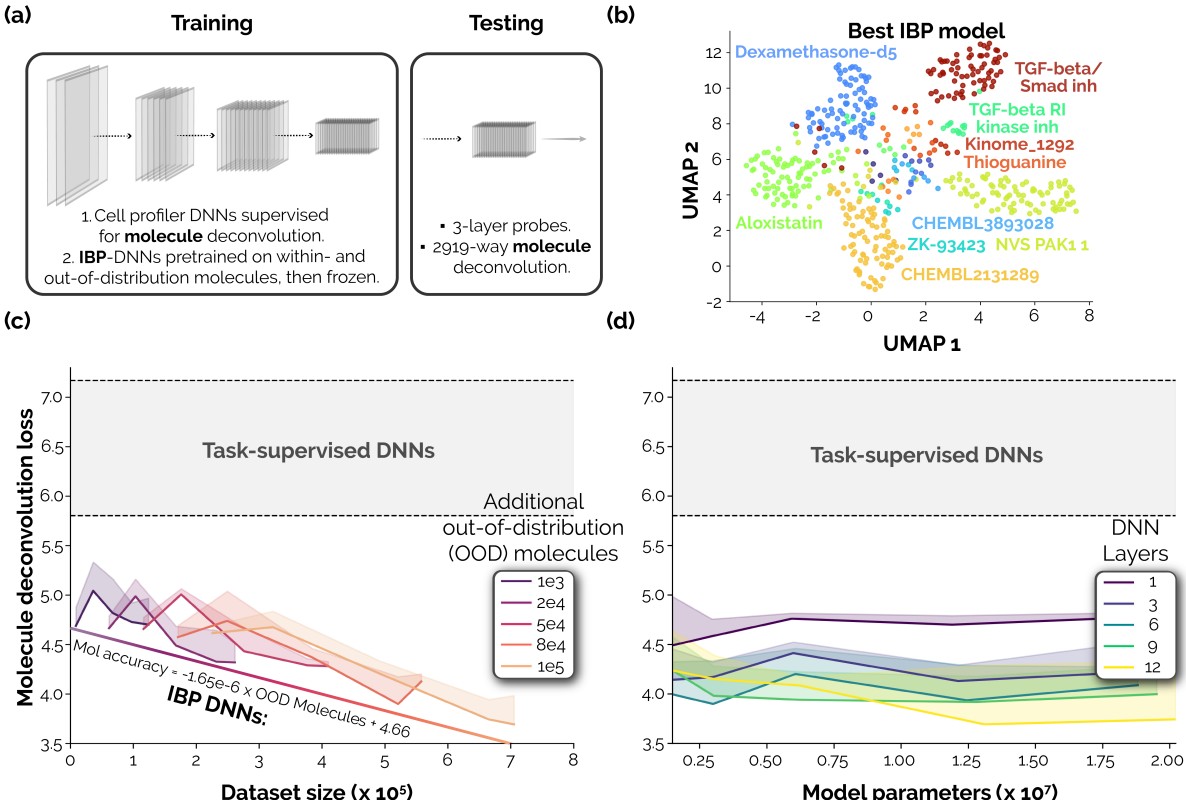

Figure A2: ***Pheno-CA challenge 3*: Molecule deconvolution.** (**a**) DNNs were either trained directly for molecule deconvolution from phenotypes or first pretrained on the IBP task then "frozen" for testing. Testing in each case involved fitting a 3-layer probe to generate target predictions for a molecule's imaged phenotype. (**b**) The highest-performing DNN was an IBP-pretrained model, and its representations discriminate between the most commonly appearing molecules. (**c**) Molecule deconvolution loss (lower is better). IBP-trained DNNs follow a scaling law, in which their performance is predicted by the number of molecules they are trained on that fall "out-of-distribution" of the *Pheno-CA*. Chance loss is 7.97. (**d**) DNN performance generally improved as models grew in depth and parameters.

