# OpenReview forum: "Neural scaling laws for phenotypic drug discovery"
_ICLR.cc/2024/Conference — ICLR 2024 Conference Withdrawn Submission_

### Official Review · Reviewer_rGf8 · 2023-10-31

**Soundness:** 1 poor
**Presentation:** 2 fair
**Contribution:** 2 fair
**Rating:** 3
**Confidence:** 4

**Summary:**

The authors describe the Phenotypic Chemistry Arena (Pheno-CA) benchmark and investigate the performance of pre-trained and randomly initialized models on four tasks related to image-based high-content screening (iHCS). They show trends in downstream task performance with increasing model and dataset sizes. They also introduce the “inverse biological process” pre-training task.

**Strengths:**

The work is very well-motivated and timely, the introduction of the IBP task is interesting, and thinking along the lines of neural scaling for iHCS data is a promising research direction.

**Weaknesses:**

The work lacks basic baselines, including evidence that would support the major claim of the unique effectiveness of the IBP pre-training task. The "neural scaling" results are not clearly presented.

**Questions:**

1. The authors repeatedly use the phrase “precursor training task” and equate this with “causal language modeling.” Causal language modeling is an example of a self-supervised training strategy or a pre-training objective. Using these more commonly used terms and removing this equivalency will improve the clarity of the work. Similarly, “task-specific readouts” refers to 3-layer MLPs / classifier heads.

2. Because the major claim of the paper relies on the novelty of the IBP task and resulting performance improvements, there should be a naive pre-training baseline other than random weight initialization. For example, prior work in NLP shows that simple synthetic pre-training tasks [Wu, Yuhuai, Felix Li, and Percy S. Liang. "Insights into pre-training via simpler synthetic tasks." Advances in Neural Information Processing Systems 35 (2022): 21844-21857.] can achieve similar benefits.

3. The authors might consider reading and citing the literature of neural scaling and “foundation models” specifically related to chemistry, where multiple prior works contain detailed discussions and results related to neural scaling and pre-training for small molecule chemistry; and [Caballero, Ethan, et al. "Broken neural scaling laws." arXiv preprint arXiv:2210.14891 (2022).], which discusses the functional form of scaling laws.

4. The axis labels in Fig 2b have no quantitative meaning and can be removed.

5. Neural scaling laws are almost always proposed with respect to a pre-training loss (e.g., cross-entropy loss for a causal language model). Improvements in this loss may or may not result in improvements measured on downstream tasks. It is confusing to refer to the MoA deconvolution accuracy results as a function of model size and dataset size as a “neural scaling law”, when there is nothing presented of the form L ~ x^(-beta). What does the IBP validation loss look like as a function of compute, model and dataset size?

6. A scaling curve provides a best-case scenario; it is not a guarantee that 100% accuracy on MoA will be achieved at 14M additional experiments. This assumption ignores aleatoric and epistemic uncertainty in the assay and resolution limitations in the scaling performance.

7. Can the authors provide more discussion and justification of why the approach uses an MLP trained on vector embeddings rather than a CNN or MLP-mixer trained directly on cell painting images?

8. What are the relevant baselines for tasks for which cell profiler is not compared against?

9. Have the authors considered regularization to prevent overfitting to their small datasets?

---

### Official Review · Reviewer_pWEQ · 2023-11-01

**Soundness:** 1 poor
**Presentation:** 1 poor
**Contribution:** 2 fair
**Rating:** 1
**Confidence:** 2

**Summary:**

This paper presents an analysis of the performance of a class of deep neural networks attacking parts of the JUMP dataset, a dataset of annotated images of painted cells (stained cell compartments). The authors study the performance of different training objectives either directly or after pretraining on a task of discovering the  molecule that led to the image. The authors find improved performance on other tasks following this pretraining.

**Strengths:**

This is an early view of an important dataset.  The pretraining task seems reasonable. The originality is limited as this is mostly an application of reasonable ideas on an existing dataset, and the implementation is relatively poor.  Unfortunately the authors try to oversell their conclusions and extrapolate based on data that spans less than an order of magnitude.

**Weaknesses:**

I was a physicist in a past life, so perhaps I am more sensitive than other audiences, because in my opinion scaling laws that extrapolate asymptotically require several orders of magnitude of variation of the inputs (think 3--5 log units, possibly more) to be convincing.  This paper hardly scratches the surface on any such respect and is clearly far from the lofty goals discussed in the intro.

Overall, the presentation is convoluted at times and makes unreasonable baseline assumptions: was the original task basically a small resnet directly predicting some of the annotations without any pretraining on any type of images?  (How should I read this sentence to understand the actual architecture: "DNNs consisted of 1, 3, 6, 9, or 12 layers of these blocks, each with 128, 256, 512, or 1512 features."?) Also what is the difference of the "Phenotypic Chemistry Arena" task that predicts the molecule ID to what the authors call inverse biological process (IBP) which is learning to predict the compound from the phenotype?

**Questions:**

The top and bottom molecule in Fig 1c are identical yet lead to different phenotypes.  The title of the figure says that different molecules yield different phenotypes in cells.  If there was an intention for some of these aspects, could you possibly clarify it?  If not, perhaps drop the second copy of the molecule.

Would any lessons from past datasets, say those released by Recursion over the years including past competitions, help with the architectures, the self-supervised tasks, or with understanding of the problem domain? Or is this JUMP dataset completely unrelated to the limited past disclosures and thus there are no lessons learned from them?

---

### Official Review · Reviewer_7a8P · 2023-11-06

**Soundness:** 2 fair
**Presentation:** 3 good
**Contribution:** 3 good
**Rating:** 5
**Confidence:** 3

**Summary:**

This paper introduces a novel inverse biological process (IBP)-trained DNN model and investigates its performance as a function of model size and data quantity. The authors observe that vanilla DNNs, which learn task-specific representations, do not follow the same scaling laws as NLP models. In contrast, IBP-trained DNNs exhibit linear scaling laws and achieve better performance improvements. The authors support their claims with experimental evidence from four drug discovery tasks. The finding of the paper regarding the scaling laws can initiate subsequent papers in drug discovery and other biological domains exploring scaling laws. However, the paper could be improved by addressing the weaknesses mentioned above.

**Strengths:**

The findings of this paper have an important implication for the field of drug discovery. To achieve better accuracy on each drug discovery task, estimating the required data quantity based on the scaling laws is critical. The neural scaling laws that have been extensively explored in the NLP domain may not be directly transferable to the biological domain. This paper provides a valuable starting point for the exploration of scaling laws in drug discovery and other biological domains.

**Weaknesses:**

- The experimental rationales are not clearly specified. This is especially problematic for readers with light domain knowledge, who may be confused about why the authors used certain experimental settings. For example, it is not clear why out-of-distribution samples are only used for the IBP-trained DNN model.
- The authors do not provide insights into why their IBP-trained DNN model exhibits linear scaling laws, while vanilla DNNs do not. This is a significant finding, and it would be helpful to understand the underlying mechanisms.

**Questions:**

Major:
- Why the IBP-trained DNN model exhibits linear scaling laws, while vanilla DNNs do not?
- Can we compare the vanilla DNN and IBP DNN with same parameter size, and same training dataset (w/ OOD molecules)?
- Providing more details about the experimental datasets and evaluation metrics.

Minor:
Page 2, Line 1: "it is can capture" -> "it can capture"
Page 3, Line 2: "drug discovery tasks" -> "drug discovery tasks."
Page 5, Chance is 7e-2 -> 7\times10^{-2}